# Microtopographic Variation in Biomass and Diversity of Living and Dead Wood in a Forest in Dongling Mountains, China

Fang Ma [1], Shunzhong Wang [2], Weiguo Sang [3,*], Shuang Zhang [1,*] and Keming Ma [1]

1   State Key Laboratory of Urban and Regional Ecology, Research Center for Eco-Environmental Sciences, Chinese Academy of Sciences, Beijing 100085, China
2   State Key Laboratory of Vegetation and Environmental Change, Institute of Botany, Chinese Academy of Sciences, Beijing 100093, China
3   College of Life and Environmental Sciences, Minzu University of China, #27 Zhongguancun South Avenue, Beijing 100081, China
*   Correspondence: swg@muc.edu.cn (W.S.); shuangzhang@rcees.ac.cn (S.Z.)

**Abstract:** Habitat heterogeneity caused by topographic variations at the local scale is the environmental basis for the establishment and evolution of biodiversity and biomass patterns. The similarities and distinctions between the effects of microtopographic variables on living wood (LWD) and dead wood (CWD) remain unknown. In the present study, the response mechanisms of biomass and species diversity patterns of living wood (LWD) and coarse woody debris (CWD) to microtopographic parameters were quantified in a warm temperate secondary forest located in Dongling Mountain, China. This quantification was achieved using a generalized additive model on a completely mapped 20-hectare permanent plot. The evaluation of biomass and species diversity of woody plants was carried out by utilizing the total basal area of all individuals and the species richness within each 20 m × 20 m quadrat as a standard. The results indicate that there are notable disparities in the influence of microtopographic elements on the LWD and CWD. In the case of LWD, microtopography accounts for 22.90% of the variation in total basal area, with convexity making a greater relative contribution than elevation, slope, and aspect. Additionally, microtopography explains 46.20% of the variation in species richness, with aspect making a greater relative contribution than elevation, convexity, and slope. Nevertheless, the influence of microtopography on CWD may only account for a deviation of 10.20% in the total basal area and 4.95% in the species richness; aspect and slope have been identified as the primary drivers in this regard. The inclusion of microtopographic factors in the model resulted in a 23.10% increase in the explanatory deviations of LWD biomass and an 8.70% increase in the explanatory deviations of CWD biomass. The findings suggest that topographic considerations have a greater impact on the biomass distribution of LWD compared to that of CWD. Conversely, the biomass of CWD is more influenced by the species richness. The presence of microtopography plays a vital role in determining the spatial distribution of species and biomass at local scales, reflecting the multiple response mechanisms and growth strategies of vegetation in response to redistribution in water, soil, and light.

**Keywords:** biodiversity; biomass; live wood; coarse woody debris; microtopography; warm temperate forest

## 1. Introduction

Forest ecosystems provide a dual role as both a carbon sink and a carbon source, making them integral components of the global carbon cycle [1]. The precise assessment of carbon stocks in forest ecosystems is of utmost importance in order to gain a comprehensive understanding of global and regional carbon budgets [2]. The evaluation of carbon storage and fluxes in forest ecosystems is made more complex by the presence of diverse constituent pools; these pools encompass live wood (LWD), soil organic matter, and woody

detritus that arise from various sources such as tree mortality, canopy damage, and pruning activities [3]. The focus of this study is on detritus in the form of coarse woody debris (CWD), since it plays a crucial role in the carbon cycle by facilitating the decomposition of live biomass and its subsequent transfer to other reservoirs, like the atmosphere or soil organic material. Given the inherent stability of soil carbon stores [4], it is possible to utilize the carbon pools of LWD and CWD as a reliable indicator for assessing the short-term dynamics of carbon in forest ecosystems. Hence, there exists a pressing necessity for further elucidation regarding the determinants of LWD and CWD biomass in forest ecosystems, as it pertains to the forthcoming global carbon budget and the management of forest resources [5]. The regulation of carbon storage in forest ecosystems has been established to be influenced by topographic and biotic variables [6,7]. Extensive evidence has been shown to establish the significance of topographical elements in shaping forest community patterns at the local scale [8]. These factors exert control over several parameters, such as light availability and soil qualities, which directly influence the development and survival of plants [9]. One illustrative instance is the influence of elevation on the distribution of tree species and its effect on biomass through the regulation of moisture and sunshine [10]. The relationship between nutrient content and temperature is commonly associated with slope and aspect. Slope plays a crucial role in determining the spatial variability of plant communities, while aspect further influences community isolation [11]. The duration for which precipitation is retained is influenced by the degree of convexity, with greater convexity resulting in improved moisture preservation [12]. The influence of microtopographic parameters can not only impact community biomass trends but also provide insights into the environmental requirements for conserving species diversity. However, despite the fact that previous studies [13–17] have extensively investigated the correlation between LWD biomass and elevation, there exists a notable vacuum in our understanding of the effect of other microtopographic parameters on LWD biomass. Furthermore, in cases where the elevation intervals are tiny at the local scale, it becomes imperative to examine the biomass of LWD in conjunction with other microtopographic parameters.

The biomass of CWD is influenced by the interplay between accumulation (disturbance, self-thinning, senescence) and depletion (decay, harvest) activities [18]. These processes are indirectly influenced by microtopography at a local scale. However, thus far, only a limited number of studies have investigated the variations in CWD biomass across different altitudes [19]. There has been a scarcity of research that incorporates additional microtopographic variables in similar studies. Meanwhile, prior investigations of the distribution of CWD biomass across different elevations have yielded inconsistent conclusions. For instance, an investigation conducted in Slovakia and Poland found a notable decrease in CWD biomass as elevation increased [20]. Conversely, another study suggests that the biomass of CWD actually increases with higher altitudes in European forests [21]. Given the information provided earlier, the relationship between CWD biomass and microtopography at a regional scale, as well as the underlying mechanisms, remain uncertain. In addition, it is worth noting that the majority of relevant research has been carried out in North America, with less emphasis given to the examination of biomass variation in Asian temperate forests affected by CWD [22].

The association between biodiversity and biomass in LWD has been a prominent subject of study in the field of ecology [23]. Numerous studies have identified positive relationships between biodiversity and biomass in forest ecosystems worldwide [24]. Nevertheless, an increasing amount of scholarly literature indicates that alterations in environmental circumstances can also influence the magnitude and quality of this association within a given system [25–28]. For instance, an increase in resource availability is known to facilitate greater biomass accumulation in the presence of more favorable environmental conditions [29]. Additionally, the relationship between diversity and productivity exhibits a unimodal link when transitioning from unfavorable to favorable conditions [30]. The impact of environmental parameters on the correlation between species richness and biomass has been relatively unexplored, mostly due to the instability of environmental

conditions [31]. Therefore, it is imperative to choose microtopographic parameters in order to evaluate the environmental influence on the correlations between biodiversity and biomass. The link between LWD mortality and CWD formation is widely acknowledged, with evidence suggesting a strong interconnection [18]. However, there is a limited body of research that has examined the potential variations in the biodiversity–biomass relationship between LWD and CWD within diverse microtopographic contexts.

The present study was carried out within a 20-hectare permanent fixed plot situated in a warm-temperate deciduous secondary forest located on Dongling Mountain in China. The analysis of spatial distribution patterns of biomass and species diversity of both LWD and CWD, in response to different microtopographic factors, was conducted using generalized additive models (GAMs), utilizing data obtained from background surveys and microtopographic data specifically collected from the aforementioned plot. We sought to determine (1) how variations in spatial patterns of biomass and biodiversity react to microtopography and the relative contributions of different microtopographic elements, and (2) whether there are similarities and dissimilarities between LWD and CWD across various microtopographic conditions.

## 2. Materials and Methods

### 2.1. Study Site

On Dongling Mountain (DLM), a permanent and fixed plot is situated within the Beijing Xiaolongmen Forest Park Reserve at coordinates 115°26′ E and 40°00′ N (Figure 1). The vegetation in this area exhibits characteristics of a warm temperate deciduous broad-leaved forest, with a relatively complicated community structure. The region has a moderately temperate continental monsoon climate characterized by the presence of four distinct seasons. The mean annual temperature is recorded at 4.8 °C, with July being the hottest month with an average temperature of 18.3 °C, and January being the coldest month with an average temperature of −10.1 °C. The duration of the frost-free period on a yearly basis is approximately 195 days, while the total number of hours of sunshine experienced throughout the year amounts to approximately 2600 h. The research area experiences an annual precipitation range of 500 to 650 mm, with the months of June and August contributing around 78% of the total precipitation. The parent soil material of mountain brown soil has been identified in previous studies [32,33].

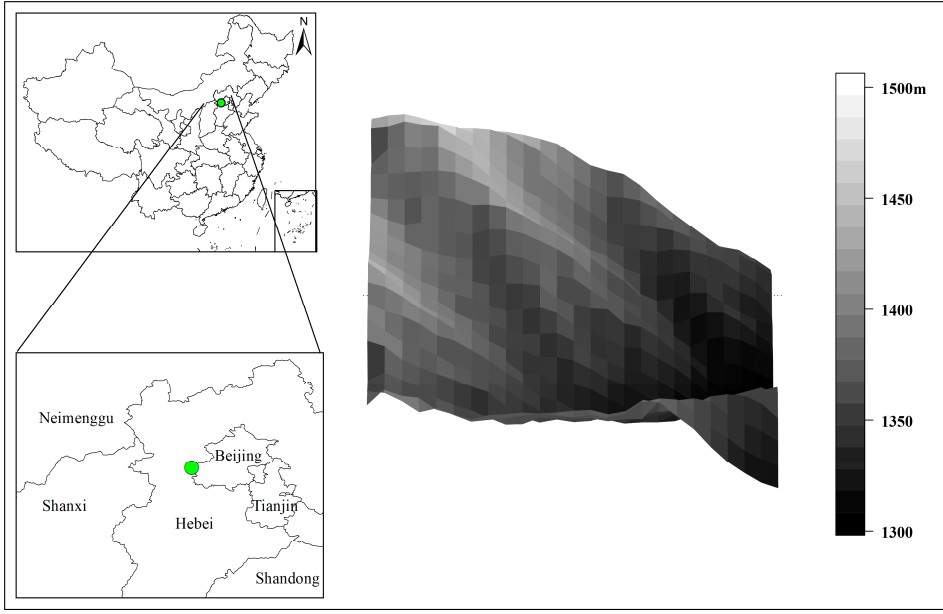

**Figure 1.** Study area and plot location.

*2.2. Data Collection*

Dongling Mountain has the characteristics of a warm-temperate broad-leaved deciduous forest, showcasing a pronounced seasonal pattern and notable stratification. The dominant tree species in this ecosystem are *Quercus wutaishanica* and many other broad-leaved deciduous trees. In earlier times, there was a high frequency of human activity, resulting in serious disturbances. In 2010, a permanent monitoring site covering 20 hectares (400 m × 500 m) was constructed on DLM. The purpose of this site is to conduct long-term investigations and conservation efforts focused on the biodiversity of warm-temperate deciduous broad-leaved secondary forest flora. The CTFS standard of using 20 m × 20 m sample plots was employed in order to assess and compare the geological and geomorphological aspects of the warm temperate forest on Dongling Mountain with other sample plots. The 20-hectare plot was partitioned into 500 subplots measuring 20*20 m each, with the origin located at the northwest corner. The recorded data for each subplot included the species, height, diameter at breast height (DBH ≥ 1 cm), and coordinates of all live woody species. The process of data collection was concluded in 2010. A comprehensive examination was conducted on a total of 500 plots, resulting in the documentation of 56 species, 36 genera, and 20 families. In 2016, a survey was undertaken to assess the post-2010 survival status of living trees. The primary objectives of this survey were to examine the coexistence patterns of living trees, analyze the mechanisms underlying tree mortality, and elucidate the interplay between living and deceased trees. The survey specifically focused on the assessment of coarse woody debris. According to the dataset from 2010, the measurements included the DBH, length or height, and coordinates of all deceased trees with a DBH greater than or equal to 5 cm. In the year 2016, a total of 32 species, 25 genera, and 15 families were documented (unidentified species as "unknown").

In this investigation, we employed the total basal area including all individuals and species richness within each 20 m × 20 m quadrat as a benchmark for assessing the magnitude of biomass and the diversity of woody plants, encompassing both living wood and coarse woody debris.

The elevation, convexity, slope, and aspect of each 20 m × 20 m quadrat were computed using elevation data acquired from the total station [34,35]. The mean elevation refers to the arithmetic mean of the elevations at the four corners of a quadrat. Convexity is determined by subtracting the mean elevation of the eight surrounding quadrats from the elevation of the focal quadrat; additionally, convexity at the edge is computed by subtracting the mean of the elevations at the four corners from the average elevation in the center of the quadrat. The quadrilateral was partitioned into four triangular facets, each of which was created by joining three vertices of the quadrilateral. The determination of the slope and aspect of each quadrat was based on the calculation of the average angle formed by the four triangular planes that deviated from both the horizontal plane and the north direction [36].

The study area under consideration classifies the shaded and sunny slopes based on the annual mean solar direction. The shaded slopes are characterized by sun orientations ranging from 0°~45° and 315°~360°, while the semi-shaded slopes encompass azimuth angles between 45°~135°. The semi-sunny slopes can be identified by sun azimuth angles ranging from 225°~315°, whereas the sunny slopes are defined by angles between 135°~225°. In total, these classifications resulted in four distinct classes. Furthermore, due to the representation of aspect within the range of 0°~360°, its direct utilization in analytical and modelling applications was limited. In cases where it is employed, mathematical conversions, such as the application of sine and cosine functions, are typically necessary to convert it into two distinct components.

*2.3. Data Analysis*

Generalized additive models (GAMs) can effectively handle the nonlinearity of independent variables in ecology, can be used to analyze data where the relationship between response variables and multiple explanatory variables is nonlinear or nonmonotonic, and

are among the most useful models for examining the connection between species resources and environmental variables [37]. GAM is a semi-parametric extension of the generalized linear model (GLM), assuming that the functions are additive and their components are smooth functions [38]. GAM uses a connection function to link the mathematical expectation of the response variable and the smooth function of the predictor variable [39].

The mathematical formula is described as follows:

$$G[E(Y)] = \beta 0 + f1(x1) + \ldots + fn(xn) \tag{1}$$

where G [ ] is the link function that depends on the response variable's distribution, E(Y) is the mathematical expectation of the response variable, β0 is the intercept term and f1,... fn is a smoothing function of n environment variables X. The smoothing spline function is the method commonly used to fit models. The identity function was chosen as the connection function, because the probability density distribution of the response variables was near normal in this study [38].

To assess the suitability of each model, the deviation coefficient ($D^2$) was calculated and indicated that the deviation represents the model's explanatory power [40]:

$$D^2 = (ND - RD)/ND \tag{2}$$

where ND is the null deviance, and RD is the residual deviance. The closer $D^2$ is to 1, the smaller the model's residual deviation and the better the model fit.

The Akaike Information Criterion (AIC) was applied to assess the goodness of fit of the model following the incremental inclusion of topographic features. According to the cited source [41], a smaller value indicates a more optimal fit for the model. The F and Chi-square tests were employed to evaluate the statistical significance of terrain factors and their non-linear impact on nonparametric effects. The GAM fitting procedure relied on the gam function, and all data analysis and visualization were conducted using the mgcv package in R version 4.1.3 (R Development Core Team, 2022).

## 3. Results

### 3.1. Effects of Microtopography on Total Basal Area

In general, the total basal area varies significantly between plots (Figure 2). For LWD (Table 1), the model's overall explained deviance was 22.9%, with convexity (12.7%), slope (6.4%), elevation (5.65%) and aspect (4.69%) in descending order. The overall basal area demonstrated a roughly linear relationship with elevation and displayed an ascending and subsequently descending pattern in relation to convexity and slope. Additionally, the basal area was shown to be higher on shaded slopes compared to sunny slopes (Figure 3). The chi-square test results revealed that the four topographic factors' non-linear contributions were all significant (Table 2). For CWD (Table 1), the model's overall explanatory deviance was 10.2%. Among the factors included, aspect contributed the most to the deviance at 4.79%, followed by convexity at 2.96%, elevation at 2.49%, and slope at 0.07%. These contributions decreased in the order mentioned (Table 1). The basal area exhibited a decline followed by an increase in response to changes in elevation. The patterns seen in convexity and aspect closely resembled those observed in LWD, whereas a positive linear relationship was observed between basal area and slope (Figure 3). Only the aspect had a significant effect on the nonlinear contribution of the nonparametric effect, according to the chi-square test (Table 2).

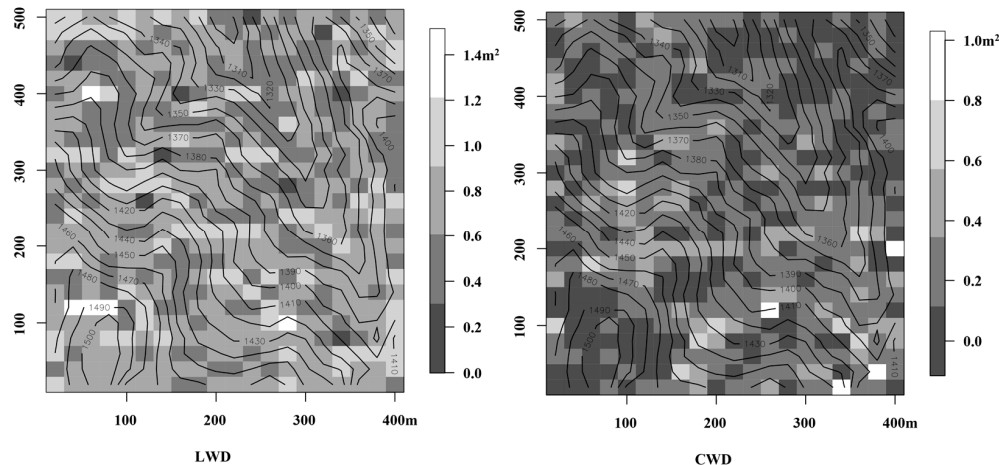

**Figure 2.** Spatial patterns of total basal area of warm forest in Dongling Mountains, Beijing.

**Table 1.** Tests of generalized additive models (GAMs) for modeling total basal area in Donglingshan and microtopographic factors.

| All Individuals | Environmental Parameters | Cumulative $R^2_{adj}$ | Cumulative Explained Deviation (%) | Akaike Information Criterion (AIC) |
|---|---|---|---|---|
| LWD | s(Elevation) | 0.053 | 5.65% (5.65%) | −170.066 |
| | s(Convexity) | 0.133 | 14.50% (12.70%) | −209.363 |
| | s(Slope) | 0.179 | 19.70% (6.40%) | −232.681 |
| | s(sin.Aspect) | 0.202 | 22.80% (4.58%) | −241.434 |
| | s(cos.Aspect) | 0.201 | 22.90% (1.55%) | −239.578 |
| CWD | s(Elevation) | 0.017 | 2.49% (2.49%) | −442.809 |
| | s(Convexity) | 0.034 | 5.51% (2.96%) | −444.669 |
| | s(Slope) | 0.033 | 5.63% (0.02%) | −443.262 |
| | s(sin.Aspect) | 0.067 | 8.24% (4.83%) | −464.660 |
| | s(cos.Aspect) | 0.076 | 10.20% (4.19%) | −463.908 |

**Table 2.** GAM model hypothesis test results of the total basal area.

| All Individuals | Smooth Terms | Edf | Ref.df | F | *p*-Value |
|---|---|---|---|---|---|
| LWD | s(Elevation) | 1.043 | 1.084 | 7.834 | 0.004 ** |
| | s(Convexity) | 4.002 | 5.008 | 7.908 | $8.12 \times 10^{-7}$ *** |
| | s(Slope) | 6.438 | 7.590 | 3.325 | 0.001 ** |
| | s(sin.Aspect) | 5.047 | 6.126 | 2.504 | 0.021 * |
| | s(cos.Aspect) | 1.000 | 1.000 | 0.023 | 0.879 |
| CWD | s(Elevation) | 4.601 | 5.681 | 2.015 | 0.061 |
| | s(Convexity) | 6.145 | 7.352 | 1.307 | 0.265 |
| | s(Slope) | 1.000 | 1.000 | 1.568 | 0.211 |
| | s(sin.Aspect) | 1.504 | 1.859 | 12.33 | $1.29 \times 10^{-5}$ *** |
| | s(cos.Aspect) | 1.000 | 1.000 | 1.301 | 0.255 |

Edf and Ref.df are the estimated degree of freedom and reference degree of freedom, respectively; "s" in "s(Elevation)" is the symbol for spline smoothing curve. *p*-value < 0.05 (*); significant difference: *p*-value < 0.01 (**); strikingly significant difference: *p*-value < 0.001 (***).

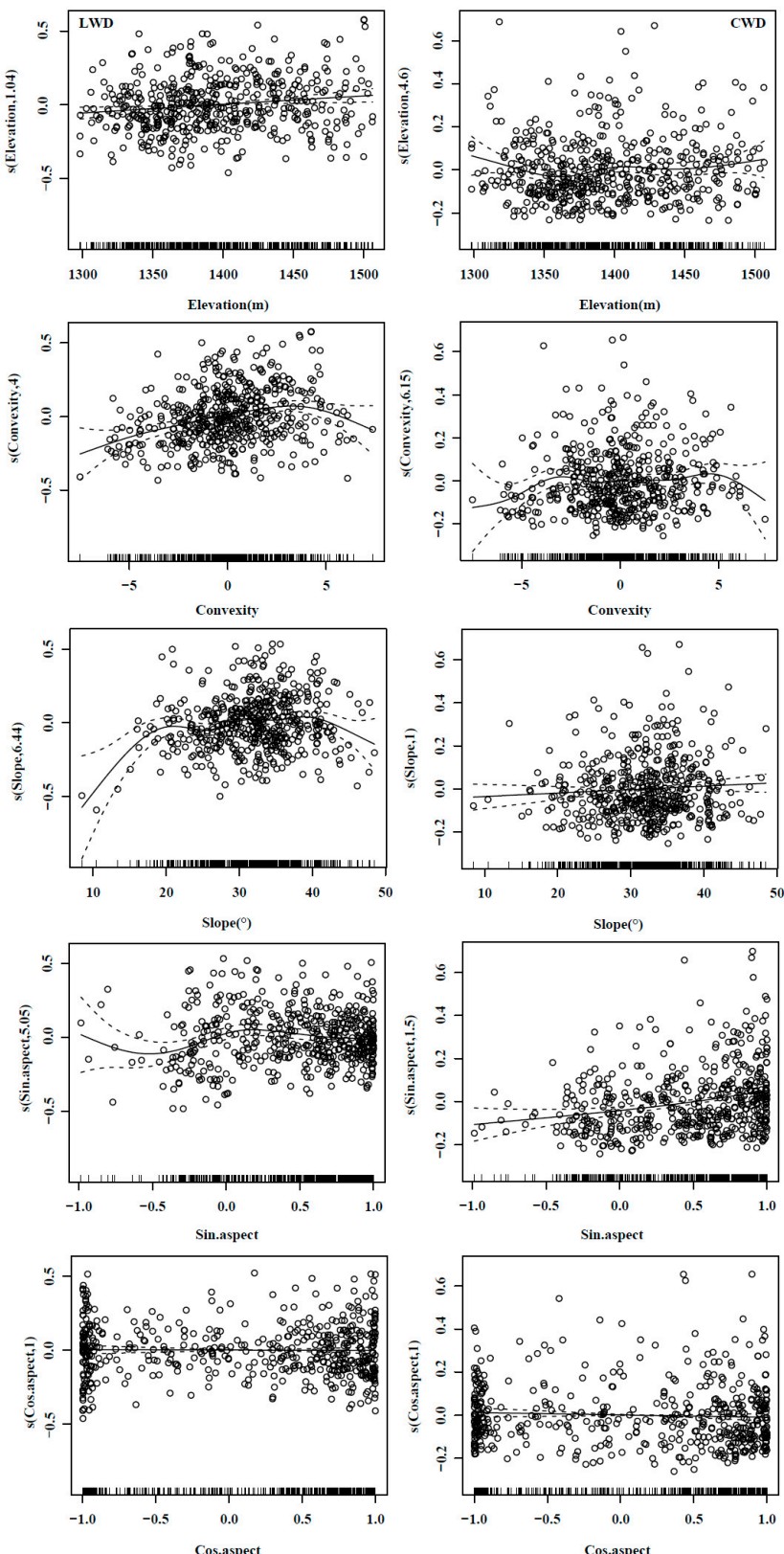

**Figure 3.** Results of generalized additive model (GAM) regression between different microtopographic factors and total basal area in Dongling Mountains, Beijing. s(microtopographic factor) is the fitted value of smoothing spline functions, which represents their impacts on the total basal area. Numbers in brackets are the estimated degree of freedom. The solid lines represent the expected values of total basal area; the dotted lines represent the 95% confidence intervals of equations. LWD means living wood, and CWD means coarse woody debris.

For LWD and CWD, variations in the estimated degrees of freedom showed that the relations between total basal area and microtopographic factors were quite different (Table 2 and Figure 3). The results of this study suggest the presence of a non-linear association between total basal area and microtopographic factors, as evidenced by the degrees of freedom (Edf, estimate degrees of freedom) for the four parameters beyond one (Table 2 and Figure 3). The degree of freedom for slope was one for CWD; the data demonstrate a positive linear relationship between the total basal area and slope (Table 2 and Figure 3).

The Generalized Cross-Validation (GCV) indicated that the performance of the fitted GAM reached its optimum at 0.036 and 0.023 for LWD and CWD (Table 3). Adjusted $R^2$ and $D^2$ showed that parts of the deviances were not explained by the fitted GAM (Table 3).

**Table 3.** Performance parameters of the fitted GAM between total basal area and microtopographic variables.

|  |  | N | Adjusted $R^2$ | $D^2$ | GCV |
|---|---|---|---|---|---|
| Model Performance | LWD | 500 | 0.201 | 0.229 | 0.036 |
|  | CWD | 500 | 0.075 | 0.102 | 0.023 |

N is the number of samples used to fit the Generalized Additive Model; $D^2$ is the deviance explained; GCV is the deviance of generalized cross-validation. Adjusted $R^2$ is the proportion of the variation in the dependent variable accounted for by the explanatory variables.

### 3.2. Effects of Microtopography on Species Richness

Significant variance in species richness can be observed throughout the various plots (Figure 4). Both LWD and CWD showed a loss in species richness as elevation increased. Additionally, they displayed a first rise followed by a subsequent fall with an increase in convexity. The species richness of LWD rose with slope, whereas CWD demonstrated the opposite pattern. The LWD species richness was higher on shaded slopes than on sunny slopes, but the difference in CWD distributions was not significant (Figure 5).

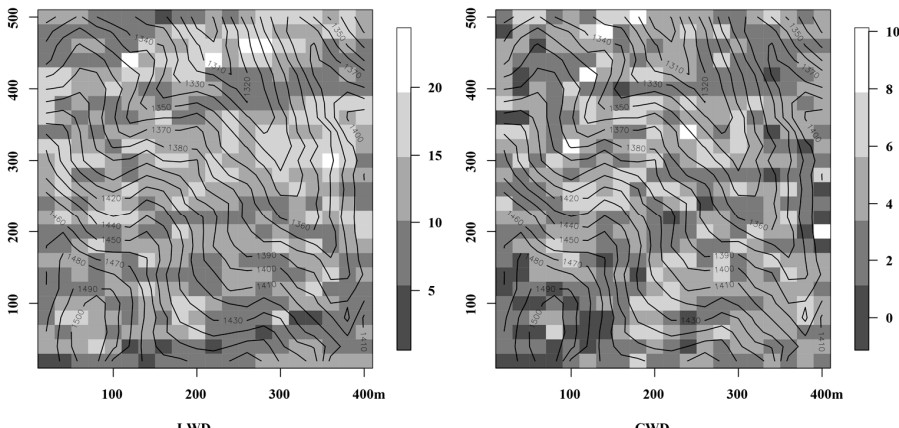

**Figure 4.** Spatial patterns of species richness of warm forest in Dongling Mountains, Beijing.

The model's total explained deviation for the species richness of LWD (Table 4) was found to be 46.20% (Table 4). Among the contributing factors, the aspect accounted for the largest proportion at 25.20%, followed by elevation at 20.10%, convexity at 10.20%, and slope at 2.78%. Based on the results of the chi-square test, it was determined that all topographic characteristics exhibited a statistically significant influence on LWD (Table 5). The model provided an explanation for the deviation seen in CWD, which amounted to 4.95% (Table 4). This deviation was attributed to various factors, with slope contributing the most at 1.77%, followed by convexity at 1.19%, aspect at 1.09%, and elevation at 0.75%, in descending order. Among the four microtopographic parameters examined, only the aspect's nonlinear contribution was shown to be statistically negligible (Table 5).

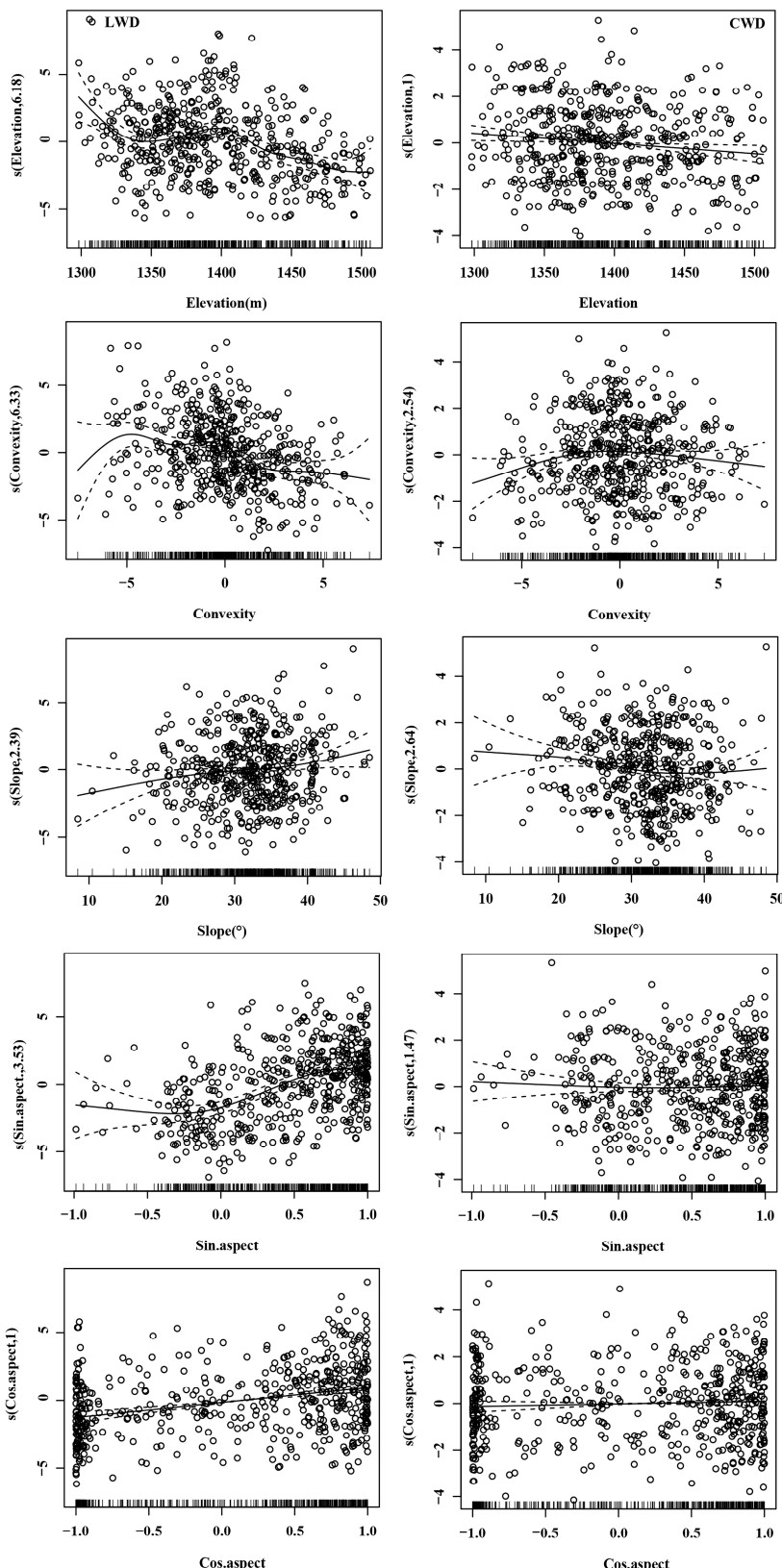

**Figure 5.** Results of generalized additive model (GAM) regression between different microtopographic factors and species richness in Dongling Mountains, Beijing. s(topographic factor) is the fitted value of smoothing spline functions, which represents their impacts on the total basal area. Numbers in brackets are the estimated degrees of freedom. The solid lines represent the expected values of total basal area; the dotted lines represent the 95% confidence intervals of equations. LWD means living wood; CWD means coarse woody debris.

**Table 4.** Tests of generalized additive models (GAM) for modeling species richness in Dongling Mountains and microtopographic factors.

| All Individuals | Environmental Parameters | Cumulative $R^2_{adj}$ | Cumulative Explained Deviation (%) | Akaike Information Criterion (AIC) |
|---|---|---|---|---|
| LWD | s(Elevation) | 0.194 | 20.10% (20.10%) | 2538.12 |
| | s(Convexity) | 0.224 | 23.20% (10.20%) | 2519.92 |
| | s(Slope) | 0.223 | 23.40% (2.78%) | 2521.53 |
| | s(sin.Aspect) | 0.394 | 41.10% (19.5%) | 2403.77 |
| | s(cos.Aspect) | 0.440 | 46.20% (22.4%) | 2369.90 |
| CWD | s(Elevation) | 0.006 | 0.75% (0.75%) | 1944.03 |
| | s(Convexity) | 0.014 | 2.14% (1.19%) | 1942.14 |
| | s(Slope) | 0.033 | 4.56% (1.77%) | 1935.09 |
| | s(sin.Aspect) | 0.032 | 4.61% (0.87%) | 1937.69 |
| | s(cos.Aspect) | 0.033 | 4.95% (0.63%) | 1936.91 |

**Table 5.** GAM model hypothesis test results of the species richness.

| All Individuals | Smooth Terms | Edf | Ref.df | F | *p*-Value |
|---|---|---|---|---|---|
| LWD | s(Elevation) | 6.184 | 7.354 | 9.693 | $<2 \times 10^{-16}$ *** |
| | S(Convexity) | 6.334 | 7.527 | 7.201 | $<2 \times 10^{-16}$ *** |
| | s(Slope) | 2.389 | 3.045 | 2.807 | 0.036 * |
| | s(sin.Aspect) | 3.529 | 4.390 | 21.60 | $<2 \times 10^{-16}$ *** |
| | s(cos.Aspect) | 1.000 | 1.000 | 40.01 | $<2 \times 10^{-16}$ *** |
| CWD | s(Elevation) | 1.000 | 1.000 | 6.815 | 0.001 ** |
| | s(Convexity) | 2.535 | 3.226 | 2.701 | 0.044 * |
| | s(Slope) | 2.643 | 3.380 | 2.585 | 0.053 |
| | s(sin.Aspect) | 1.469 | 1.805 | 0.323 | 0.703 |
| | s(cos.Aspect) | 1.000 | 1.000 | 1.111 | 0.292 |

Edf and Ref.df are the estimated degree of freedom and reference degree of freedom, respectively; "s" in "s(Elevation)" is the symbol for spline smoothing curve. *p*-value < 0.05 (*); significant difference: *p*-value < 0.01 (**); strikingly significant difference: *p*-value < 0.001 (***).

In the context of LWD, the estimated degrees of freedom were greater than one (Table 5 and Figure 5). This finding suggests a nonlinear correlation existed between species richness and microtopographic parameters, and the relationship was highly unstable. The degree of freedom for slope was one (Table 5); this indicates that there was a linear correlation between species richness and slope (Figure 5). Extrapolation by analogy, there was a linear correlation between CWD species richness and elevation (Table 5 and Figure 5). The relationship between species richness and the remaining three microtopographic variables was nonlinear (Figure 5).

According to the generalized cross-validation (GCV), the fitted generalized additive model (GAM) achieved its optimal performance at values of 6.68 and 2.81 for LWD and CWD, respectively (Table 3). The adjusted $R^2$ and $D^2$ statistics indicated that a portion of the deviances remained unexplained by the fitted generalized additive model (GAM) (Table 6).

**Table 6.** Performance parameters of the fitted GAM between species richness and microtopographic variables.

| | | N | Adjusted $R^2$ | $D^2$ | GCV |
|---|---|---|---|---|---|
| Model Performance | LWD | 500 | 0.440 | 0.462 | 6.684 |
| | CWD | 500 | 0.033 | 0.049 | 2.812 |

N is the number of samples used to fit the generalized additive model; $D^2$ is the deviance explained; GCV is the deviance of generalized cross-validation. Adjusted $R^2$ is the proportion of the variation in the dependent variable accounted for by the explanatory variables.

### 3.3. Effects of Microtopography on the Relationship of Species Richness to Total Basal Area

The percentage of total basal area explained by LWD species richness was 4.60%. However, when the topographic element was included in the model, the percentage of total basal area explained by the model increased to 27.70% (Table 7). The observed trend exhibited a transformation from an S-shaped curve to a U-shaped curve (Figure 6); this change suggests that microtopographic features have a significant role in influencing the link between species richness and biomass. The explained deviation of CWD species richness to total basal area was 20.00%; when the microtopographic element was taken into account, the explained deviation of the model to total basal area increased to 28.70% (Table 7). Furthermore, the relationship between the variables remained approximately linear (Figure 6). The influence of microtopographic parameters on LWD was shown to be more substantial compared to CWD. Additionally, the effect of CWD species richness on biomass was seen to be more significant than that of LWD, when the microtopographic parameters were not considered.

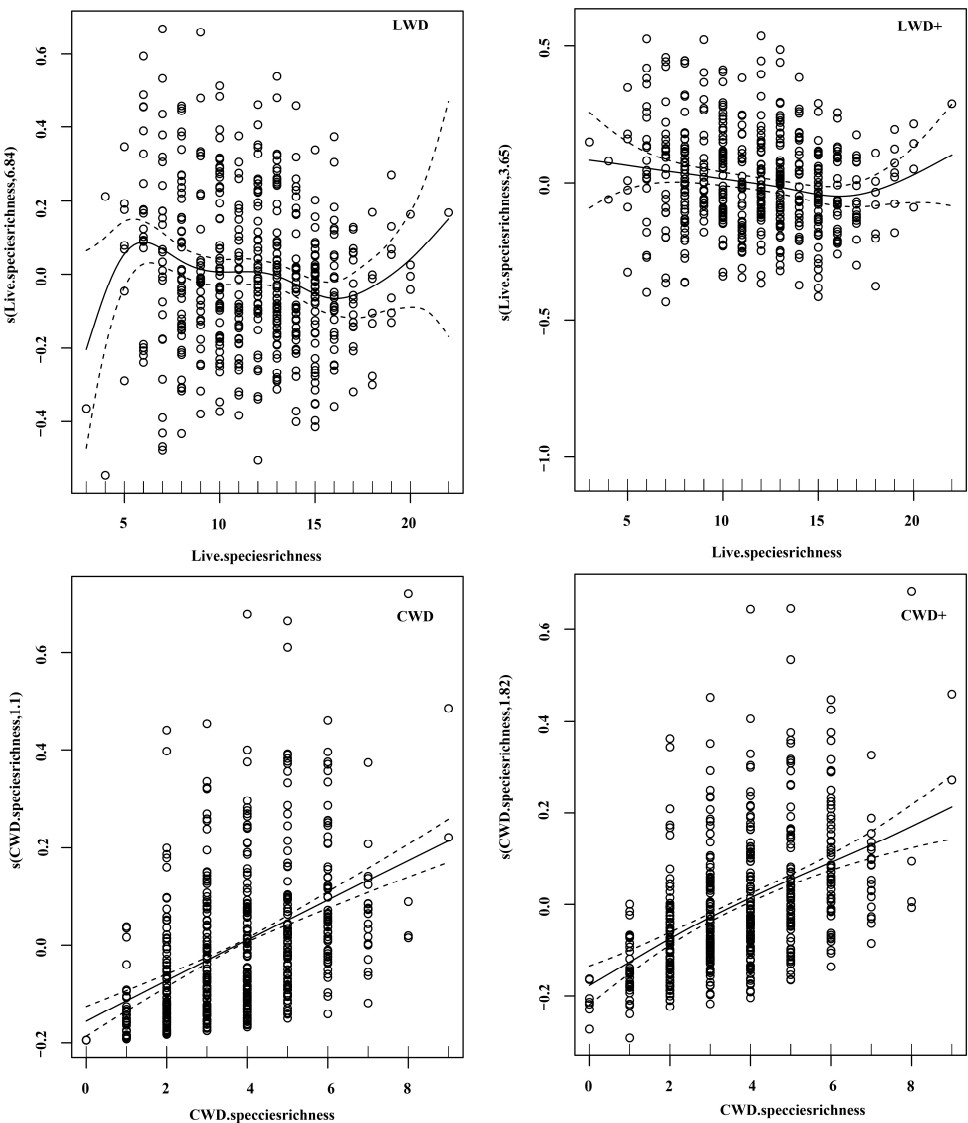

**Figure 6.** Results before and after adding microtopographic factors of generalized additive model (GAM) regression between species richness and total basal area of the warm forest in Dongling Mountains, Beijing. s(topographic factor) is the fitted value of smoothing spline functions, which represents their impacts on the total basal area. Numbers in brackets are the estimated degrees of freedom. The solid lines represent the expected values of total basal area; the dotted lines represent the 95% confidence intervals of equations.

**Table 7.** Performance parameters of the fitted GAM among species richness, total basal area and topographic variables.

| Model Performance | | Deviance Explained ($D^2$) | Adjusted $R^2$ | GCV |
|---|---|---|---|---|
| LWD | − topography | 0.046 | 0.033 | 0.043 |
| | + topography | 0.277 | 0.233 | 0.035 |
| CWD | − topography | 0.200 | 0.198 | 0.019 |
| | + topography | 0.287 | 0.271 | 0.018 |

GCV is the deviance of generalized cross-validation; smaller value means a better fit.

## 4. Discussion

### 4.1. The Response of Biomass (Total Basal Area) to Microtopography

Microtopographic elements have been extensively acknowledged as crucial determinants of forest community traits at local scales [42,43]. The relationship between microtopographic variables and plant biomass arises from the alterations in microenvironmental conditions that lead to the redistribution of hydrothermal conditions at a small spatial scale [44]. According to the findings of this research, the collective impact of microtopography on the biomass of LWD and CWD was determined to be 22.90% and 10.20%, respectively (Table 1). Our research has determined that there is a substantial association ($p < 0.05$) between LWD biomass and the following factors: convexity, slope, elevation, and aspect. The variable "aspect" demonstrated a statistically significant contribution ($p < 0.05$) to the biomass of CWD in relation to microtopography, followed by the variables "convexity", "elevation", and "slope".

Elevation has been widely recognized as a representation of moisture and light dynamics in mountain ecosystems [45], and its relationship with biomass can be characterized as positive, negative, or uncorrelated [46–48]. The findings of our study indicate a statistically significant positive correlation ($p < 0.05$) between elevation and the biomass of LWD, which follows a roughly linear trend. However, elevation exhibited the lowest contribution among the four topographic parameters. One possible explanation for this phenomenon is that a slight variation in altitude does not impose significant constraints on the growth of plants [49]. Additionally, the results of the DLM plot were in contrast with those of previous studies, highlighting the significance of geographic location and research scale [50]. The final explanation could potentially be attributed to the escalating abundance of large trees at higher elevations [46]. Previous studies have posited that the stock of CWD exhibits a decline in relation to altitude, which can be attributed to a decrease in the diameter of living wood and forest production [51,52]. The findings of this study indicate that there was a decline in CWD stocks at lower elevations, followed by an increase at higher elevations ($p > 0.05$). This pattern is consistent with previous research conducted in European forests, which also revealed a larger biomass of CWD at higher elevations [21,51]. In European forests, there is a positive correlation between altitude and the biomass of CWD, which can be attributed to the corresponding increase in the living volume of the forest. Based on our findings, the abundance of CWD is influenced by both input and output factors. The observed decrease in CWD stocks in low-elevation areas could potentially be attributed to an increase in CWD output resulting from human activities such as fuelwood extraction. Conversely, the observed increase in CWD in high-elevation areas may be attributed to a higher input of live volume from European forests, as well as a reduced output due to slower decomposition rates and less frequent disturbance.

Convexity is considered to be a highly influential element in predicting biomass [3], as it has a significant influence on moisture retention time and is correlated with soil moisture [42]. The findings of this study indicate that convexity accounted for the highest proportion (12.70%) of the variation in LWD biomass among the four parameters examined. Furthermore, the relationship between convexity and LWD biomass was statistically significant ($p < 0.05$) and demonstrated a unimodal curve. A positive association between

enough moisture and vegetation growth may exist, whereas open space is considered conducive to the growth of plants. When the level of convexity went beyond a certain threshold, it would result in a decrease in the biomass of LWD. The relationship between water allocation patterns characterized by convexity and biomass buildup is strongly correlated, as evidenced by research conducted on biomass in broad-leaf forests located in mid-latitude regions [6]. In addition, the impact of convexity on CWD biomass was shown to be statistically negligible ($p > 0.05$, Table 2). Therefore, it can be inferred that moisture does not play a significant role in tree mortality in the DLM plot. The biomass of both LWD and CWD exhibited comparable patterns of convexity, indicating a strong association between the two.

The influence of slope on various soil processes and features has been well-documented in the literature. The slope plays an essential role in driving the flow of soil water and nutrients, as well as impacting soil thickness and physicochemical characteristics [53,54], and the influence of slope on vegetation growth has been seen to be indirect in nature. The DLM plot revealed that the proportion of slope variation described by GAM with regard to LWD biomass was 6.4% ($p < 0.05$). Furthermore, our analysis indicated that steeper slopes exhibited the largest LWD biomass. According to Mascaro et al. [55], the Panamanian rainforest also exhibits a notable concentration of biomass on steep slopes. This phenomenon can be attributed to the presence of highly weathered material, which creates a favorable habitat for biomass accumulation. The quantity of living wood with big diameters on the steep slope in our plot may be attributed to the distinctive moisture and soil conditions present at this site. The biomass of CWD has a linear rise (edf = 1), and this growth occurs slowly. Several studies have revealed that forests are more vulnerable to periodic mass loss and canopy damage as slopes rise [55], and this event causes the accumulation of CWD biomass. In the context of DLM, the impact of slope on CWD biomass was not significant ($p > 0.05$). This suggests that there is less competition and mechanical damage to living wood on steeper slopes. Potential causes encompass an increase in the population of living trees and a concurrent decrease in disturbance.

Aspect is an important topographic factor in mountain ecosystems. It leads to the redistribution of water, heat, light, and soil across diverse microclimates [56]. This redistribution ultimately influences the composition of communities and the biomass of vegetation in different aspects. There are notable variations in precipitation, solar radiation, average temperature, and evapotranspiration among different aspects, leading to various plant species and developmental stages [57]. According to the DLM plot analysis, the biomass of LWD was higher on the shady and semi-shady slopes compared to those on the sunny and semi-sunny slopes. The simultaneous influence of elevated temperatures and sunshine on the sun-exposed slopes during the summer season enhances the process of transpiration and subsequent water loss, so imposing physiological limitations on the growth of LWD. In addition, the presence of wind-induced mechanical obstacles hinders the growth of trees on sun-exposed slopes, leading to a restricted accumulation of LWD biomass [49]. Furthermore, Zhu et al. found that soil organic carbon, total nitrogen, total phosphorus, available nitrogen, and available phosphorus exhibited elevated concentrations on shaded slopes compared to sunny slopes. It serves as a substantial substrate for biomass formation on shaded slopes [58]. The biomass of CWD at the DLM site had a statistically significant impact ($p < 0.05$) based on aspect. Specifically, CWD biomass was found to be higher on shady slopes than on sunny slopes. This trend in variation was consistent with the patterns of LWD. The reason may be that vegetation grows vigorously on the shaded slope, leading to intense competition among flora to produce more CWD. The accumulation of CWD, in turn, serves as a nutrient supply that facilitates the growth of LWD, thereby establishing a positive feedback loop.

### 4.2. Species Diversity (Species Richness) Response to Microtopography

The impact of microtopography on species richness within montane plant communities is widely acknowledged and represents a prominent area of interest within ecological study.

The present study observed a cumulative explained deviation of 46.20% in relation to the species richness of LWD, influenced by various microtopographic parameters. These factors, ranked in descending order of contribution, were aspect, elevation, convexity, and slope (Tables 4 and 5). The combined effect of microtopography variables explained a total variation of 4.95% in CWD species richness, with slope, convexity, aspect and elevation being the factors in order of their contribution.

In previous research on the relationship between species diversity and elevation, around 75% of the studies observed a unimodal or skewness pattern with increasing elevation [59]. Conversely, only 15% of the research found evidence of a negative effect of elevation on species diversity [60]. Studies found a significant negative relationship ($p < 0.05$) between species richness of LWD and elevation; the deviance explanation for this association was 20.1%. Firstly, the ease of colonization at lower altitudes and the subsequent acceleration of species turnover may be significant drivers. Secondly, the lower temperatures at higher altitudes may impede the process of nutrient mineralization, hence affecting ecological dynamics. Lastly, environmental filtering may play a role in the potential extinction of some species. The species richness of CWD exhibited a linear drop with elevation in the DLM plot (edf = 1, Table 5). This pattern was similar to that observed in LWD, suggesting that the decline in species richness with increasing elevation may be attributable to environmental filtering, which promotes species similarity [61].

The primary influence of convexity lies in its effect on the distribution and utilization of soil water; the habitats with smaller convexity tend to be wetter [62]. The present study elucidated that 10.2% of the species richness of LWD was accounted for by convexity, and the pattern was skewed single-peaked. The reason may be that there is a decrease in water use efficiency as convexity increases, coupled with a limited distribution of some species. The species richness of CWD also displayed a unimodal curve with convexity, maybe resulting from the combined influence of the hydrothermal environment and competition within the community.

The relationship between LWD species richness and slope was shown to be positive ($p > 0.05$), indicating that steeper slopes facilitate the colonization of a greater number of species compared to gentler slopes. This finding can be explained by the fact that trees on steeper slopes have access to more growing space and material supplies. The recorded decrease in species richness of CWD (marginally significant), was accompanied by an increase in slope. The contrasting pattern seen between LWD and CWD implies that the environmental filtering effect of the slope dimension may only exert influence on the survival and growth of a limited number of species, which are insufficient to threaten the survival and development of the majority of species.

The link between aspect and light is inherently interconnected, and light is essential for the dispersal of plants [63]. The results indicate that aspect had the greatest influence on LWD richness, explaining 25.20% of the observed variation. Additionally, species richness was found to be higher on shady (or semi-shady) slopes compared to sunny (or semi-sunny) slopes. On sunny slopes, sufficient light and heat can lead to the swift evaporation of water, resulting in reduced diversity of species. Conversely, on shady slopes, the substantial biomass contributes to a heightened input of organic matter into the soil, thereby enriching it with nutrients and fostering a higher abundance of species. The aspect variable did not have a statistically significant impact on the diversity of species in the context of CWD.

### 4.3. The Response of the Species Diversity–Biomass Relationship to Microtopography

Research conducted on natural communities has consistently demonstrated that the correlation between species richness and biomass exhibits significant variability across different ecosystems or sites, with relationships ranging from positive to negative, hump-backed, or null [64,65]. Furthermore, it has been observed that external factors, such as environmental conditions, can exert an influence on this relationship [66]. The findings of this study suggest that microtopographic variables had an impact on the relationship between species richness and biomass. Additionally, the inclusion of these variables enhanced

the model's ability to explain deviance for both LWD and CWD. The influence of abiotic factors, specifically microtopography parameters, on biomass is more pronounced in the case of LWD compared to the impact of biotic factors (species richness). Conversely, for CWD, biotic factors have a more significant effect. The findings of our study align with the observed correlation between species richness and biomass variations throughout different forest types in China, ranging from boreal to subtropical regions. However, it is important to note that the influence of species richness on biomass changes was very modest when compared to the impact of environmental conditions [30]. The research findings suggest that the significant influence of environmental factors on species richness could potentially explain the frequent occurrence of negative or inconclusive associations between diversity and biomass in natural ecosystems. This is due to the fact that the modest positive impact of species richness can be readily overshadowed by concurrent variations in environmental conditions [67]. Microtopographic parameters provide an indirect influence on the accumulation of CWD biomass by influencing the availability of LWD. Consequently, the direct effect of species richness on CWD is more significant than that of microtopography.

Furthermore, the subsequent explanations can be considered: The distribution of biomass can be influenced by the responses of different species to microtopography, including habitat preference or exclusion. For example, it has been observed that *Betula platyphylla* has a higher prevalence in ecosystems with lower elevations (unpublished). *Fraxinus rhynchophylla* serves as an extra illustration, with its CWD exhibiting a higher occurrence in valleys situated at lower elevations. The majority of species do not demonstrate either attraction or repulsion in connections with their habitats. Additionally, alterations in habitat preferences during different life stages can impact the rate of species diversity accumulation in spatial or temporal dimensions [68]. The utilization of resources at various life history stages can also impact the preferences for habitats. Fluctuations in annual precipitation cause simultaneous modifications in soil moisture and other habitat conditions, leading to notable distinctions in habitats during the initial establishment stage compared to maturity. The preference for or selection of habitats is not fixed; as forests undergo changes and life histories progress, species with limited geographic distribution may undergo a transition to become more widely distributed across various habitats. Furthermore, it is worth noting that the sample site under consideration is representative of the middle and late successional stages. This is evident from the substantial presence of juvenile tree stock and a prevailing growth-oriented species diameter pattern, suggesting a robust process of community renewal within the site. The examined site exhibits significant carbon stocks above the ground and the presence of litter serves as a nutrient substrate for the colonization of seeds, while also fostering biodiversity both above and below the ground (e.g., insect species), indicating a positive relationship between carbon stocks and biodiversity [69]. The presence of large trees facilitates bird and other animal nesting opportunities and expedites the process of seed dissemination. The senescence and mortality of large trees not only increases CWD biomass, but it also contributes to the creation of forest gaps, hence enhancing the potential for biodiversity.

Correlations were seen between microtopographic parameters in the study. Specifically, elevation presented a strong positive association with convexity and aspect, while displaying a negative relationship with slope. Additionally, convexity and slope demonstrated a significant positive correlation with aspect (Table 8). This suggests that there is a noticeable collinearity among the four microtopographic elements. Co-collinearity is a prevalent issue in the use of GAM, which should not be disregarded. Co-curvature serves as a non-parametric indicator of co-collinearity. When a model shows co-curvature, the estimated parameters may experience instability and vulnerability to the initial function [70]. The utilization of the variance inflation factor is commonly observed in practical scenarios to assess the presence of co-collinearity among predictor variables [71], and the AIC rule can be employed to determine the optimal model [72]. The objective of this study was to examine the influence of different microtopographic characteristics on the distribution of biomass and biodiversity, rather than determining the most suitable model. Furthermore,

the significance of the GAM-fitted equations was examined, and it was found that all variables exhibited statistical significance or a marginal level of significance.

**Table 8.** The Spearman correlations (*rho* values) between different topographic factors.

|           | Convexity | Slope      | Aspect  |
|-----------|-----------|------------|---------|
| Elevation | 0.35 ***  | −0.17 ***  | 0.04 *  |
| Convexity |           | 0          | 0.04 *  |
| Slope     |           |            | 0.14 ** |

\*, Significant correlation ($p < 0.05$, double-tailed test); \*\*\*, Very significant correlation ($p < 0.001$, double-tailed test). $p$-value $< 0.05$ (\*); significant difference: $p$-value $< 0.01$ (\*\*); strikingly significant difference: $p$-value $< 0.001$ (\*\*\*).

## 5. Conclusions

This study investigated the correlations between microtopographic factors, biomass, and species richness in LWD and CWD. The findings revealed that microtopographic factors had a stronger influence on LWD biomass compared to species richness. Conversely, the impact of species richness on CWD biomass was found to be greater than that of microtopography. In response to microtopography, there were both similarities and distinctions between LWD and CWD. It is worth noting that, in certain conditions, there can be indications of interdependence between the two. The observed relationship between biomass and species richness in response to microtopography has the potential to provide a basis for detecting carbon stocks and promoting biodiversity protection. Further investigation is necessary to elucidate the intricate mechanisms and establish more precise metrics for quantifying carbon sequestration in these forest ecosystems.

**Author Contributions:** Conceptualization, methodology, validation, formal analysis, investigation, writing—original draft preparation, F.M.; resources, data curation, S.W.; writing—review and editing, supervision, S.Z.; Resources, Supervision, K.M.; Resources, Supervision, W.S. All authors have read and agreed to the published version of the manuscript.

**Funding:** This research received no external funding.

**Data Availability Statement:** The data in this study are available from the authors upon request.

**Conflicts of Interest:** The authors declare no conflict of interest.

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
