# Peer review of "Microtopographic Variation in Biomass and Diversity of Living and Dead Wood in a Forest in Dongling Mountains, China"

_forests, doi:10.3390/f14102111_

Round 1

Reviewer 1 Report (Previous Reviewer 2)

The authors addressed most of the previously reported problems. The quality of the work is much improved, and relevant content has been added. Therefore, I think that the admission of the work in the new form can be considered.

I ask the authors to check the English for minor mistakes. Minor changes can still be made in the document.

Author Response

Dear reviewer:

Thank you for your comments concerning our manuscript entitled “Microtopographic variation in biomass and diversity of living and dead wood in forests from in forest from Dongling Mountains, China”. These comments are all valuable and very helpful for revising and improving our paper, as well as the important guiding significance to our researches. We have studied comments carefully and have made corrections which we hope meet with approval.

Thank you again for your positive and constructive comments and suggestions on our manuscript.

We hope you will find our revised manuscript acceptable for publication.

Reviewer 2 Report (Previous Reviewer 1)

The Paler was significant improved. However, should be considerated some aspects:

- the statistical approach should bebrevised. In sucul cases of multiple parameters , a complementary în vestigation methods as PCA, FA, etc.  In this sau,  the significance of each parameters could be revealed.

- the quality of approaching method should not be  redused to a single quality result parameter. The R square, R square arbust, etc should be included to. The significance coefficient should be also included.

- the main observation îs related to forest composition. In many papers, the forest composition and the age distribution were presented in order to prezent the initial  state of study. In fact, for prezent paper these specific aspecte are not presented. In this respect, for such Little are of 20 km squares, se could not avere with the Paler conclusion statele for entire geografical ardă as North China.

In the best approaching way, the study could be redused for this particular investigated area.

Author Response

Dear reviewer:

We would like to thank you for your careful reading, helpful comments, and constructive suggestions, which has significantly improved the presentation of our manuscript.

We have carefully considered all comments from the reviewers and revised our manuscript accordingly. The manuscript has also been double-checked, and the typos and grammar errors we found have been corrected. In the following section, we summarize our responses to each comment from the reviewers. We believe that our responses have well addressed all concerns from the reviewers. We hope our revised manuscript can be accepted for publication.

Comment:

- the statistical approach should be b revised. In sucul cases of multiple parameters , a complementary în vestigation methods as PCA, FA, etc.  In this sau,  the significance of each parameters could be revealed.

- the quality of approaching method should not be  redused to a single quality result parameter. The R square, R square arbust, etc should be included to. The significance coefficient should be also included.

- the main observation îs related to forest composition. In many papers, the forest composition and the age distribution were presented in order to prezent the initial  state of study. In fact, for prezent paper these specific aspecte are not presented. In this respect, for such Little are of 20 km squares, se could not avere with the Paler conclusion statele for entire geografical ardă as North China.

In the best approaching way, the study could be redused for this particular investigated area.

Reply:

  1. Principal Component Analysis (PCA) is commonly employed to reduce the dimensionality of various environmental components. However, in the present study, only four indicators are utilized to represent the environmental factors. Consequently, the application of dimensionality reduction techniques, such as PCA, may not yield substantial significant outcomes. The proportional importance of each element has been delineated in Table 1 and Table 4, where the cumulative explained deviation (%) is provided. The rate of each factor was elucidated, with the respective values provided in parenthesis. Furthermore, the objective of this study is to investigate the correlation between microtopography, biomass, and biodiversity, with a particular focus on the overall significance of topographic factors and the trade-offs between abiotic and biotic factors in determining biomass. This will be achieved by analyzing the simulation outcomes of the Generalized Additive Model (GAM), rather than solely determining the most suitable model. This particular topic will be discussed in detail in future work.

Table 1. Tests of generalized additive models (GAM) for modeling total basal area in Donglingshan  and microtopographic factors

All individuals

Environmental parameters

Cumulative

R2adj

Cumulative explained deviation (%)

Akaike Information
Criterion (AIC)

LWD

s(Elevation)

0.053

5.65% (5.65%)

-170.066

s(Convexity)

0.133

14.50% (12.70%)

-209.363

s(Slope)

0.179

19.70% (6.40%)

-232.681

s(sin.Aspect)

0.202

22.80% (4.58%)

-241.434

s(cos.Aspect)

0.201

22.90% (1.55%)

-239.578

CWD

s(Elevation)

0.017

2.49% (2.49%)

-442.809

s(Convexity)

0.034

5.51% (2.96%)

-444.669

s(Slope)

0.033

5.63% (0.02%)

-443.262

s(sin.Aspect)

0.067

8.24% (4.83%)

-464.660

s(cos.Aspect)

0.076

10.20% (4.19%)

-463.908

Table 4. Tests of generalized additive models (GAM) for modeling species richness in Donglingshanand microtopographic factors

All individuals

Environmental parameters

Cumulative

R2adj

Cumulative explained deviation(%)

Akaike Information
Criterion(AIC)

LWD

s(Elevation)

0.194

20.10% (20.10%)

2538.12

s(Convexity)

0.224

23.20% (10.20%)

2519.92

s(Slope)

0.223

23.40% (2.78%)

2521.53

s(sin.Aspect)

0.394

41.10% (19.5%)

2403.77

s(cos.Aspect)

0.440

46.20% (22.4%)

2369.90

CWD

s(Elevation)

0.006

0.75% (0.75%)

1944.03

s(Convexity)

0.014

2.14% (1.19%)

1942.14

s(Slope)

0.033

4.56% (1.77%)

1935.09

s(sin.Aspect)

0.032

4.61% (0.87%)

1937.69

s(cos.Aspect)

0.033

4.95% (0.63%)

1936.91

  1. The cumulative R-square findings, as shown in Table 1 and Table 4, include the R-square values. This is done since the study of the single R-square, particularly in the CWD results, reveals a little gap. On the other hand, the deviance explained results provide a more explicit understanding. The tables (Table 2, Table 5) also include the coefficient of significance, commonly referred to as the P-value.

Table 2. Approximate significance of smooth terms

All individuals

Smooth Terms

Edf

Ref.df

F

p-Value

LWD

s(Elevation)

1.043

1.084

7.834

0.004**

s(Convexity)

4.002

5.008

7.908

8.12e-07***

s(Slope)

6.438

7.590

3.325

0.001**

s(sin.Aspect)

5.047

6.126

2.504

0.021*

s(cos.Aspect)

1.000

1.000

0.023

0.879

CWD

s(Elevation)

4.601 

5.681

2.015

0.061.

s(Convexity)

6.145  

7.352

1.307

0.265

s(Slope)

1.000  

1.000

1.568

0.211

s(sin.Aspect)

1.504 

1.859

12.33 

1.29e-05***

s(cos.Aspect)

1.000

1.000

1.301

0.255

Table 5. Approximate significance of smooth terms

All individuals

Smooth Terms

Edf

Ref.df

F

p-Value

LWD

s(Elevation)

6.184  

7.354 

9.693

<2e-16 ***

S(Convexity)

6.334  

7.527 

7.201

<2e-16 ***

s(Slope)

2.389 

3.045 

2.807 

0.036 * 

s(sin.Aspect)

3.529   

4.390

21.60

<2e-16 ***

s(cos.Aspect)

1.000

1.000

40.01

<2e-16 ***

CWD

s(Elevation)

1.000  

1.000

6.815 

0.001 **

s(Convexity)

2.535   

3.226

2.701

0.044*

s(Slope)

2.643   

3.380

2.585

0.053.

s(sin.Aspect)

1.469   

1.805

0.323

0.703 

s(cos.Aspect)

1.000

1.000

1.111

0.292

Thank you again for your positive and constructive comments and suggestions on our manuscript. We hope you will find our revised manuscript acceptable for publication.

Round 2

Reviewer 2 Report (Previous Reviewer 1)

We still consider that the article could be improved.
The study on a relatively small area and using only one analysis method, can be considered - but not always - sufficient.
the quality of the analysis and results -
It says: GAM is a semi-parametric extension of generalized linear model (GLM), assuming that the functions are additive and their components are smooth functions.
But what results would a GLM type model give? In a GLM model, the effect size can be easily obtained and the classification of the influence can be done easily.
in fact, table 6 specifies the values for R square Adjusted - values that actually say that the obtained model is weak.
nor are the graphs in figures 3 and 5 more explicit.
I think that the work could be improved.

Author Response

Dear reviewer:

Thank you very much for the positive comments and constructive suggestions. Please find the following detailed responses to your comments and suggestions.

Response:

  1. The rationale for using generalized additive models (GAM) is in their efficacy in addressing the nonlinearity of independent variables in the field of ecology. GAMs have been recognized as one of the useful models for investigating the correlation between species resources and environmental variables (Guisan et al., 2002).
  2. In this study, our primary emphasis was placed on examining the impact of environmental factors on total basal area, also known as biomass. To illustrate the influence of specific components, we have presented the corresponding trends in Figure 3 and Figure 5. Due to the interdependence among the components, it is more advantageous to examine them collectively rather than focusing solely on the isolated effects of individual factors, which are regarded as signs of changing patterns.
  3. Table 6 illustrates the correlation between topographic factors and richness. Our primary focus is on discerning the pattern of this association, rather than establishing a specific model. Additionally, the P-value may offer a more comprehensible measure in comparison to R2.
  4. In Figure 3 and Figure 5, the vertical coordinate reflects the dependent variable. The value indicates the estimated degree of freedom and is a fixed output pattern for the GAM model, but it can be removed if necessary. The depicted images serve to illustrate the impact of the independent variable, such as elevation, on the dependent variable, namely the basal area. In relation to the aspect of picture clarity, it has been observed that the submitted images are of high resolution and in vector format.
  5. In light of the singular nature of the data, it is imperative to integrate it with additional datasets in order to facilitate a more comprehensive analysis in further research.

We would like to thank the referee again for taking the time to review our manuscript.

This manuscript is a resubmission of an earlier submission. The following is a list of the peer review reports and author responses from that submission.

Round 1

Reviewer 1 Report

The article is interesting and the topic is original. Unfortunately, the presentation is not well structured and neither are the working hypotheses. basically, an area of ​​20 hectares was considered and conclusions were considered for the entire area of ​​Dongling Mountain. From the title it appears that this study covers the entire temperate zone of China! Punctual observations 1. The purpose of the work should be better explained 2. the research methodology must be exposed more clearly. DIn the article it appears that in 2010 plots were demarcated and then in 2016 comparisons were made. 3. I don't think that such a method could be accepted without specifying more details that would give weight to the study carried out, which is certainly interesting 4. Figure 1 is not very explicit and there are no location maps - probably because the study was intended to be valid for an extended area 5. The statistical methods used are those often found in the literature, but it lists a statistically descriptive part of the measured elements. 6. Biodiversity is mentioned in the article, but nothing is mentioned about the way the biodiversity analysis was done, nor are the dominant species specified, or at least 2-3 more common species are listed. 7. For this reason, the fact that the influence of surface curvature and the influence of slopes was analyzed on a relatively small surface without presenting the entire set of parameters that were taken into account makes me consider that this study requires substantial changes. 8. It does not appear anywhere what the measured parameters are and whether they are considered blank surfaces and/or previous records. 9. The included graphs (fig. 3, fig. 4 and fig. 5) are of poor quality and cannot be studied As a result, I recommend major revision  

Author Response

Dear reviewer:

Thank you for your decision and constructive comments on my manuscript. We have carefully considered the suggestion of Reviewer and make some changes. We have tried our best to improve and made some changes in the manuscript.

The red part that has been revised according to your comments. Revision notes, point-to-point, are given as follows:

Comment 1:

Title: From the title it appears that this study covers the entire temperate zone of China! Punctual observations

Response 1:

We think this is an excellent suggestion. We have revised the title “Microtopographic variation in biomass and diversity of living and dead wood in forest from Dongling Mountains, China”.

Comment 2:

The purpose of the work should be better explained

Response 2:

We examined: 1)How spatial patterns of biomass and biodiversity vary under the in-fluence of microtopography and the relative contributions of various microtopographic factors? 2)Whether there are similarities and dissimilarities between LWD and CWD under various microtopographic conditions?

 P2, Line 95-98.

Comment 3:

the research methodology must be exposed more clearly. D In the article it appears that in 2010 plots were demarcated and then in 2016 comparisons were made.

Response 3:

Thanks for your suggestion.

Dongling Mountain is a warm-temperate broad-leaved deciduous forest with a dis-tinct seasonal character and significant strata, dominated by Quercus wutaishanica and other broad-leaved deciduous trees. Human activities were frequent and disturbances were severe in the past. A 20-hectare (400m*500m) permanent monitoring site of warm-temperate deciduous broad-leaved secondary forest was established on DLM in 2010 to investigate and protect the biodiversity of vegetation over the long term. Based on the geological and geomorphological characteristics of the warm temperate forest in the Dongling Mountain, as well as in order to compare the results with those of other sample plots, the CTFS standard of 20 m*20 m sample plots was utilized. The 20-ha plot was di-vided into 20* 20 m subplots with the northwest corner as the origin, yielding 500 subplots. All live woody' species, height, diameter at breast height (DBH≥1 cm), and coordinates were recorded for each subplot. In 2010, data collection was completed. A total of 500 plots were investigated, and 56 species, 36 genera and 20 families were recorded. In order to include the survival status of living trees (seedlings, juvenile trees, etc.) after 2010, to investigate the coexistence of living trees, to analyze the mechanism of tree mortality, and to ex-plain the relationship between living and dead trees, a survey of coarse woody debris was conducted in 2016. Based on the 2010 data set, we added DBH, length, and coordinates of any dead trees with DBH≥5 cm in 2016, 32 species, 25 genera and 15 families were recorded.

P3-4 , Line 113-132.

Comment 4:

Methods

I don't think that such a method could be accepted without specifying more details that would give weight to the study carried out, which is certainly interesting.

The statistical methods used are those often found in the literature, but it lists a statistically descriptive part of the measured elements.

Response 4:

Thank you for pointing this out.

The reviewer is correct, and we have added Table1-6 in the result.

Comment 5:

Figure 1 is not very explicit and there are no location maps - probably because the study was intended to be valid for an extended area

Response 5:

Thanks for your suggestion. 

We have added a new map. P3,Line 109.

We

Comment 6:

Biodiversity is mentioned in the article, but nothing is mentioned about the way the biodiversity analysis was done, nor are the dominant species specified, or at least 2-3 more common species are listed.

Response 6:

Thank you for pointing this out. We have added content on diversity.

P2-3, Line 124,129. Fig 4.

Fig 4. Spatial patterns of species richness of warm forest in Donglingshan ,Beijing

Comment 7:

For this reason, the fact that the influence of surface curvature and the influence of slopes was analyzed on a relatively small surface without presenting the entire set of parameters that were taken into account makes me consider that this study requires substantial changes.

Response 7:

Thank you for the detailed review.

The sample sites are established in accordance with CTFS criteria and supplement the global monitoring network for biodiversity. Numerous investigations have been conducted using the CTFS monitoring network as a foundation. The above was added to 2.2 Data collection.

P3, Line 112-120.

For example,

Lai, J.; Mi, X.; Ren, H.; Ma, K. Species-habitat associations change in a subtropical forest of china. J Veg Sci. 2009, 20, 415-23.

Comment 8:

It does not appear anywhere what the measured parameters are and whether they are considered blank surfaces and/or previous records.

Response 8:

Thanks for your constructive suggestion.

This section is mainly located at 2.2 Data collection

P3, Line 111-142.

Comment 9:

The included graphs (fig. 3, fig. 4 and fig. 5) are of poor quality and cannot be studied As a result, I recommend major revision .

Response 9:

All of the images have been edited and added. In addition, to assure the quality of the images, I will simultaneously upload vector images.

We tried our best to improve the manuscript and made some changes marked in red in revised paper which will not influence the content and framework of the paper. We appreciate for Reviewers’ warm work earnestly, and hope the correction will meet with approval. Once again, thank you very much for your comments and suggestions.

Sincerely,

The Authors

Reviewer 2 Report

Dear authors,

The objective of this study was to shed light on the particularities between the effects of microtopographic variables on the living wood and dead wood using a generalised additive model.

In my opinion, the paper fits the scope of the Forest journal, and the analysed topic is relevant and interesting. I also believe the paper can make a solid contribution to the literature. The well-structured manuscript shows challenging work (e.g. a fully mapped 20-ha permanent plot). It also has a clear methodology (for data collection - from the DBH, length, and coordinates of each tree, to obtaining the convexity and slope from the elevation data; and data analysis – describing the functions of the generalised additive model) although it lacks some in-depth analyses and more finishing touches. However, I will recommend its publication, with major changes, considering the importance of this study in determining the structure of secondary forests in China.

Some personal suggestions:

Title: Microtopographic variation in biomass and diversity of living and dead wood in warm temperate forest in China

The study can not be generalised to China, so the authors should be more specific in the title: for example, “… in forests from Dongling Mountain, China”.

In the abstract – “We used the total basal area of all individuals and species richness in each 20 m × 20 m quadrat as standard for measuring the value of biomass and species diversity of woody plants.

Measuring should be replaced by assessing.

The final phrase: “The importance of microtopography on species and biomass distribution at the local scale, reflecting the multiple response mechanisms and growth strategies of vegetation influenced by the redistribution of soil, water, and light conditions.” has to be revised because the sentence lacks the predicate.

1. The introduction is clear and succinct, focusing on the microtopographic factors. It was emphasised the objective of the study and the fact that few researchers have assessed the biodiversity-biomass relationship between the living and dead wood across varied microtopography conditions.

2 Methods

Fig 1 – maybe also a regional map should be added for information.

Also some additional info:

- on the selection of that particular 20-ha plot. What are its dimensions (ratio between length and width)? (is not explicitly specified in section 2, although after analysing Fig. 2 from the results, I figured out it was 500x400m).

- summary of the forest structure and maybe the history of activities in the studied area 

- additional explanations should be given regarding the continuity of the data sets - the 2010 set and the 2016 set (because dimensional differences could occur).

- on the 20x20 quadrats. What was the reason for that division, and how was their size chosen? Is this scale relevant enough to capture the differences? Can the authors elaborate on their choice using similar examples from the bibliography?

- for data collection – what was the resolution for the elevation data obtained from the total station?

- microtopography parameter integration into the model (in 2.3 2.3 Data Analysis the GMA is generally presented)

3 Results

I think the authors presented the relevant information on the GAM tests and the performance parameters of the fitted GAM.

4 Discussion

The results and their interpretation are exciting, considering the study examined the relationship among microtopographic factors, biomass and species richness. However, I consider that knowing the species composition of the forest is relevant, which could bring more understanding of the results and their interpretations. Furthermore, it could be relevant for how the species react to varied microtopography conditions, especially considering the biomass of LWD.

I also consider particularly interesting the dissociation between the microtopographic factors (which influenced living wood bio-mass more than species richness) and species richness (which was more relevant for dead wood biomass). However, maybe, for discussion, the authors could also read and include more recent studies – for example (and no, I am not an author of that particular study ;-)) 

Mikoláš, M., Svitok, M., Bače, R., Meigs, G. W., Keeton, W. S., Keith, H., ... & Svoboda, M. (2021). Natural disturbance impacts on trade-offs and co-benefits of forest biodiversity and carbon. Proceedings of the Royal Society B, 288(1961), 20211631.

In this study, the aboveground carbon storage was higher in old-growth forest development stages, particularly in large stems and dead wood sites, with an increased biodiversity potential.

Please, consider improving the discussions by focusing on this aspect.

5. Conclusions.

I suggest revising it for the small mistakes.

The English

A smoother English is recommended – many minor faults:

Revised abstract (the last phrase).

line 44 "a urgent" …an urgent

line 70 “CWD increases with altitude in European.” … maybe in European forests or stands.

line 75 “with positive biodiversity-biomass correlations are found in forests” … “are” should be eliminated

and some other small mistakes.

A smoother English is recommended – many minor faults:

Revised abstract (the last phrase).

line 44 "a urgent" …an urgent

line 70 “CWD increases with altitude in European.” … maybe in European forests or stands.

line 75 “with positive biodiversity-biomass correlations are found in forests” … “are” should be eliminated

and some other small mistakes.

Author Response

Dear reviewer:

Thank you for your decision and constructive comments on my manuscript. We have carefully considered the suggestion of Reviewer and make some changes. We have tried our best to improve and made some changes in the manuscript.

The red part that has been revised according to your comments. Revision notes, point-to-point, are given as follows:

Comment 1:

Title: Microtopographic variation in biomass and diversity of living and dead wood in warm temperate forest in China

The study can not be generalised to China, so the authors should be more specific in the title: for example, “… in forests from Dongling Mountain, China”.

Response 1:

We think this is an excellent suggestion. We have revised the title “Microtopographic variation in biomass and diversity of living and dead wood in forest from Dongling Mountains, China”.

Comment 2:

In the abstract – “We used the total basal area of all individuals and species richness in each 20 m × 20 m quadrat as standard for measuring the value of biomass and species diversity of woody plants.

Measuring should be replaced by assessing.

The final phrase: “The importance of microtopography on species and biomass distribution at the local scale, reflecting the multiple response mechanisms and growth strategies of vegetation influenced by the redistribution of soil, water, and light conditions.” has to be revised because the sentence lacks the predicate.

Response 2:

We sincerely thank the reviewer for careful reading. We have revised this part according to the Reviewer’s suggestion.

“measuring was replaced by assessing.” P1, Line 17.

we have rewritten this sentence as follows:

“Microtopography is crucial for the distribution of species and biomass at local scales, reflecting the multiple response mechanisms and growth strategies of vegetation in response to redistribution in water, soil, and light.” P1, Line 27-29.

Comment 3:

Fig 1 – maybe also a regional map should be added for information.

Response 3:

Thanks for your suggestion.

We have added a new map. P3,Line 109.

Comment 4:

Methods

Also some additional info:

- on the selection of that particular 20-ha plot. What are its dimensions (ratio between length and width)? (is not explicitly specified in section 2, although after analysing Fig. 2 from the results, I figured out it was 500x400m).  P3, Line 115

- summary of the forest structure and maybe the history of activities in the studied area. P3, Line112-116

- additional explanations should be given regarding the continuity of the data sets - the 2010 set and the 2016 set (because dimensional differences could occur).  P3-4, Line124-128

- on the 20x20 quadrats. What was the reason for that division, and how was their size chosen? Is this scale relevant enough to capture the differences? Can the authors elaborate on their choice using similar examples from the bibliography?  P3, Line117-120

- for data collection – what was the resolution for the elevation data obtained from the total station? 5m

- microtopography parameter integration into the model (in 2.3 2.3 Data Analysis the GMA is generally presented)

Response 4:

Thank you for pointing this out. The reviewer is correct, and we have modified. The revised text reads as follows.

Dongling Mountain is a warm-temperate broad-leaved deciduous forest with a distinct seasonal character and significant strata, dominated by Quercus wutaishanica and other broad-leaved deciduous trees. Human activities were frequent and disturbances were severe in the past. A 20-hectare (400m*500m) permanent monitoring site of warm-temperate deciduous broad-leaved secondary forest was established on DLM in 2010 to investigate and protect the biodiversity of vegetation over the long term. Based on the geological and geomorphological characteristics of the warm temperate forest in the Dongling Mountain, as well as in order to compare the results with those of other sample plots, the CTFS standard of 20 m*20 m sample plots was utilized.

In order to include the survival status of living trees (seedlings, juvenile trees, etc.) after 2010, to investigate the coexistence of living trees, to analyze the mechanism of tree mortality, and to explain the relationship between living and dead trees, a survey of coarse woody debris was conducted in 2016.

Regarding the last point, it would have been interesting to explore this aspect, detailed explanations of all parameters are provided in the results.

Comment 5:

Discussion

The results and their interpretation are exciting, considering the study examined the relationship among microtopographic factors, biomass and species richness. However, I consider that knowing the species composition of the forest is relevant, which could bring more understanding of the results and their interpretations. Furthermore, it could be relevant for how the species react to varied microtopography conditions, especially considering the biomass of LWD.

I also consider particularly interesting the dissociation between the microtopographic factors (which influenced living wood bio-mass more than species richness) and species richness (which was more relevant for dead wood biomass). However, maybe, for discussion, the authors could also read and include more recent studies – for example (and no, I am not an author of that particular study ;-)) 

Mikoláš, M., Svitok, M., Bače, R., Meigs, G. W., Keeton, W. S., Keith, H., ... & Svoboda, M. (2021). Natural disturbance impacts on trade-offs and co-benefits of forest biodiversity and carbon. Proceedings of the Royal Society B, 288(1961), 20211631.

In this study, the aboveground carbon storage was higher in old-growth forest development stages, particularly in large stems and dead wood sites, with an increased biodiversity potential.

Please, consider improving the discussions by focusing on this aspect.

Response 5:

We think this is an excellent suggestion.

To improve the article's readability and substance, we modified the discussion in accordance with the reviewers' comments.

P14, Line 420-442

We sincerely appreciate the valuable comments. We have checked the literature carefully and added more references on 70 and 71 into the discussion part in the revised manuscript.

Additionally, the following are possible explanations: Firstly, the distribution of biomass can be affected by the responses of various species to microtopography, such as habitat preference or exclusion. For instance, Betula platyphylla is more common in low-elevation habitats (other research that I have not published). Fraxinus rhynchophylla is another example; its CWD is more prevalent in low-elevation valleys. The majority of species exhibit neither attraction nor repulsion in their habitat associations. Secondly, changes in habitat preferences at various life stages may influence the rate at which species diversity accumulates in space or time [70]. Light resource use at different life history stages can also influence habitat preferences; interannual variation in precipitation causes concurrent changes in soil moisture and other habitat conditions, resulting in significantly different habitats at the young tree establishment stage versus the maturity stage. Habitat preference or selection is not static; as forests evolve and life histories develop, a unique habitat preference can evolve into adaptations to other habitats, thereby expanding the distribution of narrow-range species to include wide-range species. Thirdly, the sample site belongs to the middle and late successional stages, with a large amount of juvenile tree stock and a predominantly growth-oriented species diameter pattern, indicating healthy community renewal. The sample site has high above-ground carbon stocks, and the litter not only provides nutrient substrates for seed colonization but also promotes above- and below-ground biodiversity (e.g., insects, etc.) and increases CWD biomass, establishing a positive correlation between carbon stocks and biodiversity [71]. Large trees provide nesting locations for birds and other animals, as well as speed up seed dispersal. Not only do the senescence and death of large trees increase the CWD biomass, but they also create forest apertures, thereby increasing the potential for biodiversity.

Comment 6:

Fig 1 – maybe also a regional map should be added for information.

Response 6:

Revised abstract (the last phrase).

line 44 "a urgent" …an urgent

line 70 “CWD increases with altitude in European.” … maybe in European forests or stands.

line 75 “with positive biodiversity-biomass correlations are found in forests” … “are” should be eliminated

and some other small mistakes.

We feel sorry for our carelessness. In our resubmitted manuscript, the typo is revised. Thanks for your correction.

Thank you again for your positive comments and valuable suggestions to improve the quality of our manuscript. If there are any other modifications we could make, we would like very much to modify them and we really appreciate your help. Thank you very much for your help.

Sincerely,

The Authors

Round 2

Reviewer 2 Report

I thank the authors for their effort. All the issues I raised have been clarified and corrected adequately by the authors.